# Factors Influencing Ground Settlement during Tunnel Proximity Construction

Xun Yuan [1], Hongchao Wang [1], Shun Kang [2,*], Changwu Liu [3], Yulin Chen [3], Xianliang Zhou [2], Chengzhe Wu [4], Haowei Zhu [5], Changyu Yang [1], Yong Zhu [1] and Hua Wu [1]

[1] China Railway Eryuan Engineering Group Co., Ltd., Chengdu 610031, China; yuanxun3@ey.crec.cn (X.Y.); wanghongchao@ey.crec.cn (H.W.); yangcy@ey.crec.cn (C.Y.); zhuyong@ey.crec.cn (Y.Z.); wuhua@ey.crec.cn (H.W.)
[2] School of Emergency Management, Xihua University, Chengdu 610039, China; xianliangzhou@stu.scu.edu.cn
[3] College of Water Resource & Hydropower, Sichuan University, Chengdu 610065, China; yulinchen@stu.scu.edu.cn (Y.C.)
[4] China Chengda Engineering Co., Ltd., Chengdu 610096, China; chengzhewu@stu.scu.edu.cn
[5] Sichuan Highway Planning, Survey, Design and Research Institute Ltd., Chengdu 610041, China; haoweizhu@stu.scu.edu.cn
*   Correspondence: kangshun5683@163.com

**Abstract:** Adjacent tunnel excavation has an adverse impact on existing structures. Based on the engineering project of the Donghuashan Tunnel under-crossing an existing tunnel, this paper designed 25 sets of orthogonal numerical simulation tests to investigate the influential mechanisms of five parameters on ground displacement and deformation. The influential factors are skew angle ($\alpha$), proximity distance ($l$), buried depth ($h$), clearance ($D$), and ratio of tunnel clearances ($\nu$). The orthogonal test results revealed that (1) the new tunnel clearance is the main impact factor of both ground settlement and curvature deformation, (2) ground horizontal movement is most significantly influenced by the skew angle between the existing tunnel and the new tunnel, and (3) the new tunnel buried depth is the key influential parameter for ground tilt deformation as well as horizontal deformation. The conclusions of this research suggest that during the period of railway planning, it is very important to plan the buried depths and spans of new tunnels rationally to minimize disturbance to existing tunnels.

**Keywords:** adjacent tunneling; under-crossing excavation; influential parameter; ground settlement; ground deformation





## 1. Introduction

Numerous railway tunnel constructions take place as social demand for transportation increases [1], in which the safety of existing structures (tunnels, stations, and buildings) adjacent to the tunnels are a main concern. Tunnel construction entails the creation of ground settlements, which can endanger the adjacent buildings when tunnels under-pass, superimpose, or side-pass through the existing structures. The responses of existing structures and influencing mechanisms of impact factors therefore need to be investigated thoroughly.

Previous scholars have investigated the problems faced in adjacent tunneling engineering, such as the Zhaotong tunnel of Yukun high-speed railway. Cheng et al. [2] defined three safety factors that can quantify the serviceability limit state of existing tunnels, and this enabled a quick assessment of the excavation-induced tunnel damage potential. Their research subject was subway tunnels in urban areas. Daniela et al. [3] investigated the response of framed buildings to tunneling, and the results were summarized in terms of the deflection ratios and modification factors for horizontal strains. A satisfactory agreement between predictions and measurements was obtained. Over-crossing tunneling adversely affect, and can even damage, existing tunnels if the induced deformation exceeds the design

limit of tunnel structures. Liu et al. [4] proposed a new model for evaluating the behaviors of underlying tunnels prior to construction. The impact factors on existing tunnels such as advancing distance, clearance distance, and the stiffness of joints were also investigated through their research. Tunneling may also induce excessive internal forces and displacements of adjacent building piles, emphasizing the necessity of predicting the pile responses during the preliminary tunnel design. Mohammad et al. [5] studied the parameters of face pressure, grout pressure, and thrust force in a driving tunnel to investigate their effects on ground movement and tunnel–underpass interactions. Chen et al. [6] considered the spatial variabilities of rock mass properties when investigating the under-crossing tunneling problem. Lai et al. [7] explored the settlement characteristics of an existing tunnel caused by under-crossing tunneling in close proximity with a low intersection angle. They revealed that the vertical settlement and torsional deformation are the main types of deformation of the existing tunnel caused by tunneling underneath. Lee et al. [8] took anisotropic and time-dependent behaviors into account in order to estimate the deformation of tunnel excavation in slate formation. Tunneling may also have an adverse effect on existing jointed pipelines due to the induced ground movement. Taking this into account, Huang et al. [9] proposed an improved Winkler solution to predict the pipelines' responses. Huang et al. [10] investigated the failure mechanism of the surrounding rocks of a tunnel induced by adjacent excavation. They found that the compact lining structure of the existing tunnel may fail due to the construction of a new tunnel and constructed a new failure mechanism. Based on this, they derived the upper-bound solution of the slip surface equation for the rock mass around an existing tunnel in the framework of the upper-bound theorem in conjunction with a variational approach.

Laboratory experiments present visual phenomena arising from tunneling problems. Mukhtiar et al. [11] conducted a series of centrifuge model tests to investigate the effects of the construction sequence of twin stacked tunnel advancement on an existing pile group under working load, revealing that tunnel construction sequences had substantial effects on pile group settlement, pile cap tilting, and lateral movement in the pile group. They also investigated the load transfer of pile due to tunnel construction [12]. The effects of tunnel construction on ground movements in sand were determined by Sohaei et al. [13] through a series of experiments. They found that the settlement trough width increased almost linearly with increments of the overburden. It could be concluded that the weight of existing structures above the excavating tunnel intensifies the ground movement, thus destroying the structure itself.

Numerical simulation methodology has also been widely adopted to explore the adjacent tunneling problem. Mojtaba et al. [14] developed three-dimensional numerical analyses to study the interaction between twin tunnels and underground parking, construction of underground parking above the existing tunnels and excavation of the twin tunnels under the existing parking, revealing the effect of construction sequence on the soil and the structures' behavior. Ground settlement influence area and horizontal strain distribution are vital impact factors regarding building damage. Anna et al. [15] simulated and investigated the effect of raft foundation on a pre-existing tunnel through PLAXIS software. Li et al. [16] used ADINA software to simulate the effects of tunnel construction on the settlements of the ground surface and pile foundations in the composite strata. Johannes et al. [17] presented a benchmark of a rate-dependent constitutive model for soft soils, implemented in a 2D finite element code, compared to the response of an instrumented excavation in sensitive clay. The results of modeling agree with the ongoing settlement rate assessed by remote sensing data. Zhang et al. [18] established a three-dimensional simulation model to investigate the ground response to twin-tunnel construction. Yao et al. [19] investigated the use of isolation piles during metro tunnel construction to protect adjacent buildings, utilizing the method of finite difference software FLAC3D. A numerical method was also adopted by Prateep et al. [20] to investigate tunnel deformation due to adjacent loaded pile and pile–soil–tunnel interactions. They recommended an assessment method for tunnel deformation, defined by the maximum extension and maximum contraction of

the tunnel diameter and their associated axes with respect to the horizontal and vertical directions, respectively. Li and Zhang [21] employed an anisotropic soil constitutive model NGI-ADP in a finite element simulation to investigate pile responses (deflection, bending moment, and shaft resistance) to tunneling. Considering excavation-induced disturbance to the surrounding soil, Liu et al. [22] utilized a Timoshenko beam to simulate a shield tunnel. Li et al. [23] constructed a three-dimensional model to simulate dynamic responses around an existing tunnel under unloading disturbance forces.

Moreover, twin-tunnel construction faces the same problems as those produced by adjacent excavation. The mobilization of shear displacement of the strata between the two tunnels may lead to an increase in their persistence and a reduction in shear strength, as well as a weakening of the whole rock mass of the middle wall. As part of a case study, LiDAR was utilized to determine the emergency during the construction of the twin-bore outlet tunnels at the Zengwen Reservoir in Taiwan [24]. According to the Analysis of Controlled Deformations (ADECO) principles, the Cassia twin road tunnels under-crossing Cassia road in Italy were full-face excavated to 260 m$^2$ [25]. This kind of construction allowed for full control of the ground ahead of the face. Using field measurement, Peng et al. [26] analyzed the effect of double-line parallel shield tunneling on the deformation of adjacent buildings. They highlighted the pass effect of excavation on buildings after the tunnel construction is finished.

However, the construction of building structures will also disturb the stability of existing tunnels. During construction, tunnels and buildings can be influenced by each other. The disturbance is, in fact, mutual. In soft soils, large-scale excavation may exert a great influence on nearby tunnels. Displacements, cracks, and leakages of a metro tunnel in Ningbo were observed by Chen et al. [27] through field monitoring. Based on monitoring data, the responses of the ground and tunnel to the adjacent excavation were investigated. Liang et al. [28] considered both the bending effect and the shearing deformation of the tunnel to predict its longitudinal responses to adjacent excavations. The tunnel–ground interaction was considered by introducing a two-parameter Pasternak foundation, which could further take account of the interaction between adjacent springs. Soil unloading in foundation pit engineering can adversely affect tunnels in the vicinity. Adjacent excavation during pile constructions inevitably changes ground stress state and leads to soil movements around nearby tunnels. Therefore, exploring the responses of existing shield tunnels associated with adjacent excavation tunnels is crucial and essential. Liang et al. [29] introduced the Pasternak foundation model with a modified subgrade modulus to predict shield tunnel behaviors associated with adjacent excavation. Zhang et al. [30] proposed a simplified analytical approach to explore the deformation response of adjacent tunnels to excavation-induced soil-unloading in excavation engineering. Zhang et al. [31] proposed a semi-analytical method to evaluate the heave of an underlying tunnel induced by adjacent excavation, obtaining the influence of excavation and the resistance of tunnel through Boussinesq's and Mindlin's solutions, respectively.

There are also studies on building protection methodologies. The adjacent construction of a tunnel may contribute to possible damage to or operational safety concerns for existing buildings. Therefore, proper measures must be taken. Underground cut-off wall, grouting reinforcement technique, and an optimized construction parameter were adopted, combined with field monitoring, to protect adjacent buildings from the Bund Tunnel construction in Shanghai [32]. Considering the long-term performance of tunnels, Liu et al. [33] proposed micro-disturbance grouting to correct the deformed tunnel influenced by adjacent excavation. The methodologies of a large-diameter pipe screen, settlement measurement system, and in-pipe grouting system were utilized during the under-crossing construction of the twin tunnels under an operating airport runway [34]. The project was finished without any interruption to the runway.

Despite the research progress outlined above, a research gap exists: the relationship between the impact parameters of skew angle, proximity distance, buried depth, clearance, ratio of tunnel clearances (see Table 1 and Figure 1), and ground displacement and defor-

mation is still unclear. To narrow this research gap in the field of adjacent tunnel excavation engineering, the impact mechanisms of five influential factors on ground settlement were investigated, using the Donghuashan Tunnel project as an example.

**Table 1.** Ground subsidence impact factors.

| Level | Factors | | | | |
|---|---|---|---|---|---|
| | Skew angle/° ($\alpha$) | Proximity distance/m ($l$) | Buried depth/m ($h$) | Clearance/m ($D$) | Ratio of tunnel clearances ($v$) |

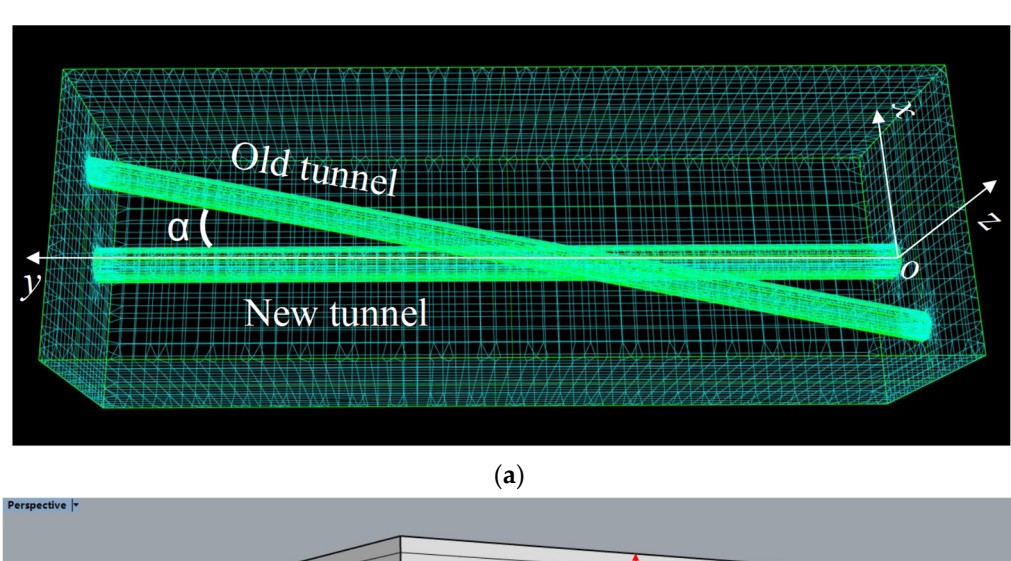

(**a**)

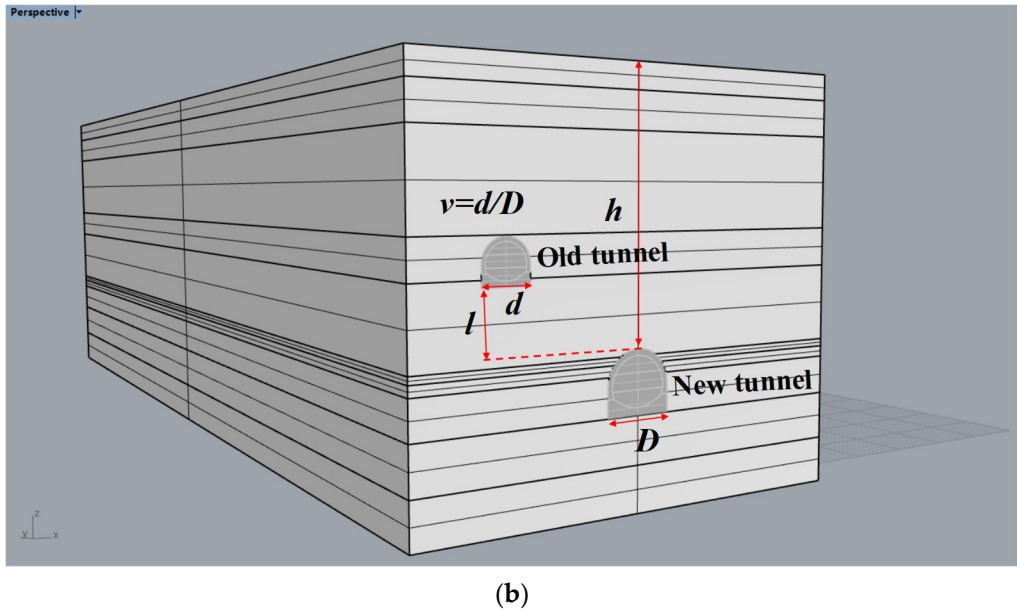

(**b**)

**Figure 1.** Relative position of old and new tunnel. (**a**) Top view, (**b**) front view.

## 2. Materials and Approach

### 2.1. Project Overview

Figures 2 and 3 show the location and top view of the existing Yangjiagou Tunnel and the new Donghuashan Tunnel. The Donghuashan Tunnel, which lies on the Hanbanan Railway, is now under construction in Bazhong, a small city in west China.

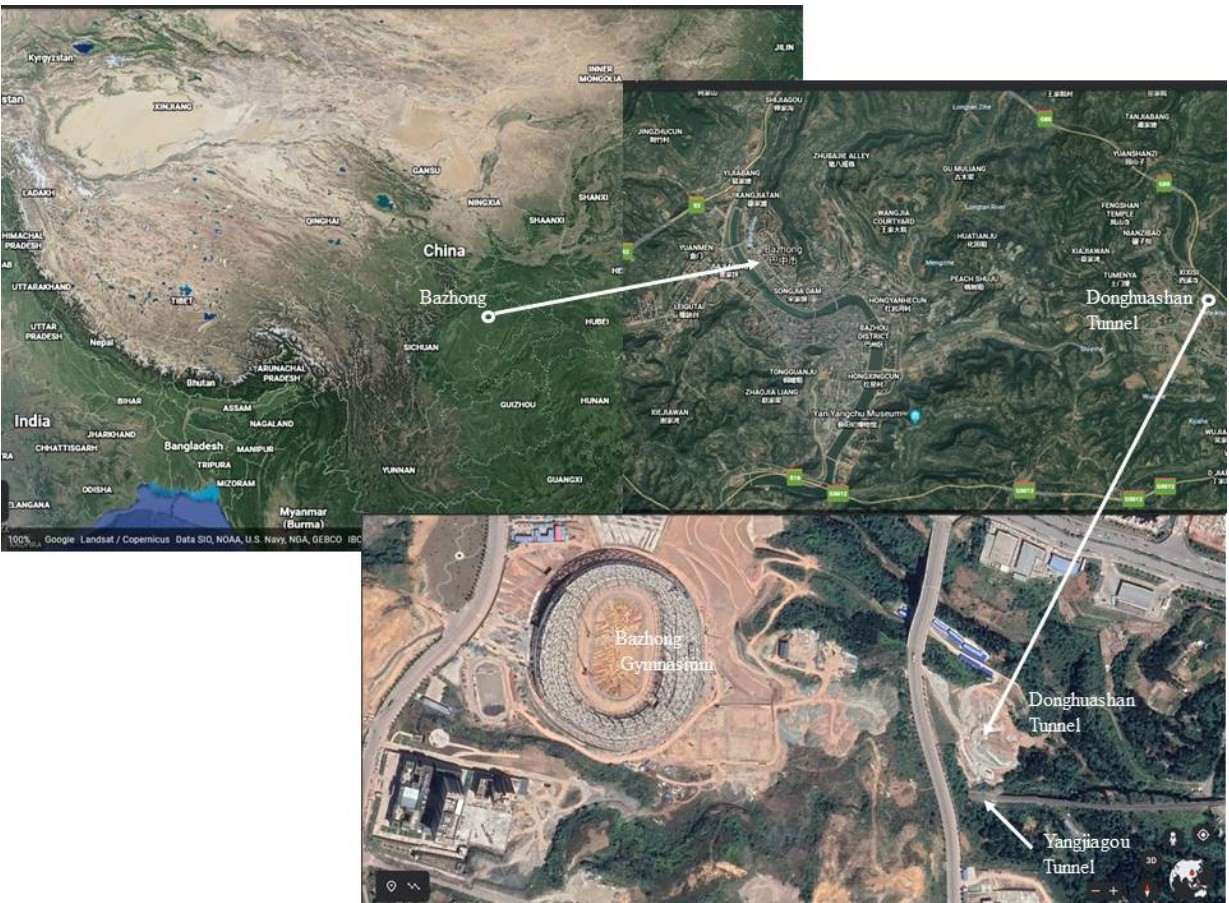

**Figure 2.** Location and top view of the Donghuashan Tunnel (provided by Google Earth).

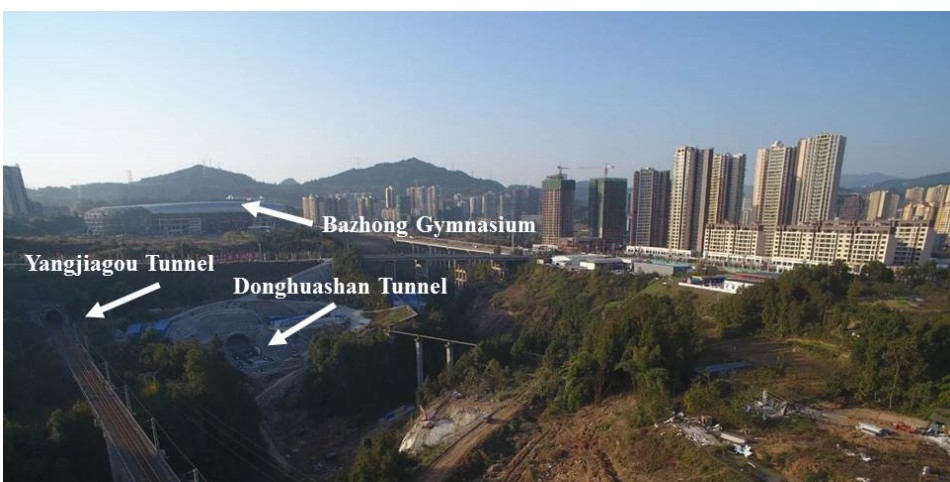

**Figure 3.** Position of Donghuashan Tunnel relative to the existing structures.

The new tunnel was designed to under-cross the existing tunnel at a skew angle of 11.3° and the Bazhong Gymnasium due to the limitations of the construction site. The minimum proximity distance of the two tunnels in the intersection zone was only 13.5 m. The existing Yangjiagou Tunnel has been under construction since January 2016, while the Bazhong Gymnasium is still under construction; therefore, their stability is a priority during the construction of Donghuashan Tunnel. The topsoil is artificial accumulation, followed by highly weathered sandstone, sandstone, siltstone, and conglomerate. Table 2 lists the mechanical parameters of the strata. The under-crossing tunnel was constructed

using the method of milling excavation for the purpose of controlling the subsidence of overlying strata and the stability of the existing tunnel. The clearance of the new tunnel is 14.52 m with a lining thickness of 0.75 m. The buried depth of the new tunnel is 56.58 m.

**Table 2.** Mechanical parameters of the strata.

| No. | Strata | $h$ | $\rho$ | $\varphi$ | $c$ | $Rt$ | $E$ | $\mu$ |
|---|---|---|---|---|---|---|---|---|
| | | m | Kg/m³ | ° | MPa | MPa | GPa | |
| ①$_1$Q$_4$$^{ml}$II | Artificial accumulation | 5.6 | 1939 | 33 | 0.024 | 0.01 | 0.03 | 0.29 |
| ⑨$_{12}$K$_{1b}$$^{Ss}$IVW$_3$ | Highly weathered sandstone | 7.9 | 2245 | 43 | 0.03 | 0.14 | 0.75 | 0.15 |
| ⑨$_{13}$K$_{1b}$$^{Ss}$IVW$_2$ | Sandstone | 19.9 | 2347 | 35 | 0.7 | 4.9 | 6 | 0.3 |
| ⑨$_{33}$K$_{1b}$$^{St}$IVW$_2$ | Siltstone | 8.2 | 2296 | 39 | 0.5 | 3.6 | 5 | 0.31 |
| ⑨$_{13}$K$_{1b}$$^{Ss}$IVW$_2$ | Sandstone | 16.4 | 2143 | 35 | 0.7 | 4.9 | 6 | 0.3 |
| ⑨$_{73}$K$_{1b}$$^{Cg}$IVW$_2$ | Conglomerate | 1.6 | 2347 | 35 | 0.7 | 4.9 | 6 | 0.3 |
| ⑨$_{33}$K$_{1b}$$^{St}$IVW$_2$ | Siltstone | 2.3 | 2296 | 39 | 0.5 | 3.6 | 5 | 0.31 |
| ⑨$_{13}$K$_{1b}$$^{Ss}$IVW$_2$ | Sandstone | 8.2 | 2143 | 30 | 0.2 | 2 | 3.8 | 0.32 |
| ⑨$_{33}$K$_{1b}$$^{St}$IVW$_2$ | Siltstone | 10.1 | 2296 | 39 | 0.5 | 3.6 | 5 | 0.31 |

Note: $h$ is the thickness of the strata; $\rho$ is the density of the rock mass; $\varphi$ is the internal friction of the rock mass; $c$ is cohesion; $Rt$ is the uniaxial compressive strength of the intact rock material; $E$ is the rock mass modulus of elasticity; $\mu$ is the Poisson's ratio of rock mass.

## 2.2. Orthogonal Test Design

Here, we designed five levels of orthogonal array ($L_{25}(5^6)$) to investigate the influence of the five main influential parameters on ground subsidence as well as the stability of the existing tunnel under the condition of under-crossing tunneling. The mentioned influential parameters are the skew angle of the two tunnels, their proximity (the center-to-center spacing between the two tunnels), the buried depth of the newly excavated tunnel, the new tunnel's clearance, and the ratio of the two tunnels' clearances. The levels of each geometrical parameter are listed in Table 3.

**Table 3.** Orthogonal test design schemes.

| Level | Factors | | | | |
|---|---|---|---|---|---|
| | Skew Angle/° ($\alpha$) | Proximity Distance/m ($l$) | Buried Depth/m ($h$) | Clearance/m ($D$) | Ratio of Tunnel Clearances ($v$) |
| 1 | 11.3 | 13.5 | 56.58 | 14.52 | 0.7 |
| 2 | 30 | 10 | 50 | 12 | 0.8 |
| 3 | 50 | 15 | 60 | 14 | 1.0 |
| 4 | 70 | 20 | 70 | 16 | 0.6 |
| 5 | 90 | 25 | 80 | 18 | 0.5 |

## 2.3. Simulation Model

According to engineering data of Donghuashan Tunnel, the numerical simulation model was designed with the geometry of length × width × height = 300 m × 100 m × 70.1 m, comprising 1,456,580 zones and 259,139 grids (Figure 4). Figure 5 shows the relative position of the new tunnel to be excavated and the existing tunnel.

conglomerate1
conglomerate2
donghuashan1
donghuashan2
donghuashan3
floor
floor2
plainfill
sandstone1
sandstone2
sandstone3
sandstone4
sandstone5
shell
shell2
siltstone1
siltstone2
siltstone3
siltstone4
weatheredsandstone
yangjiagou1
yangjiagou2
yangjiagou3

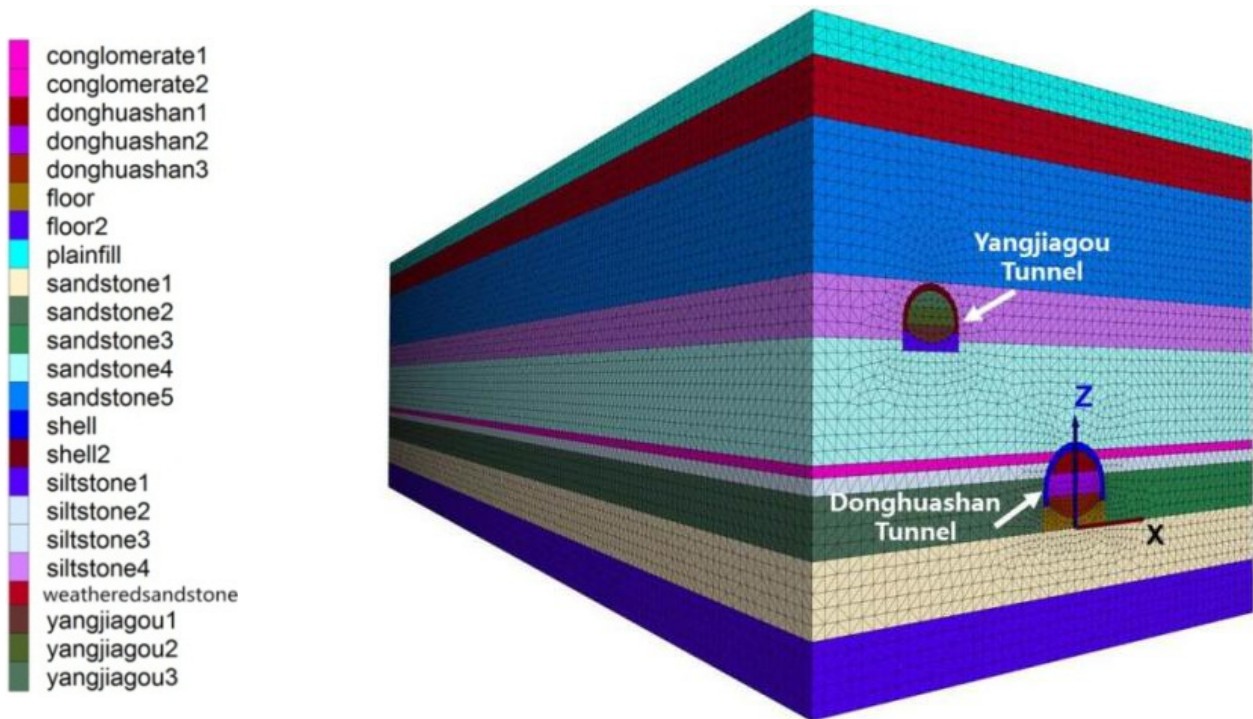

**Figure 4.** Simulation model.

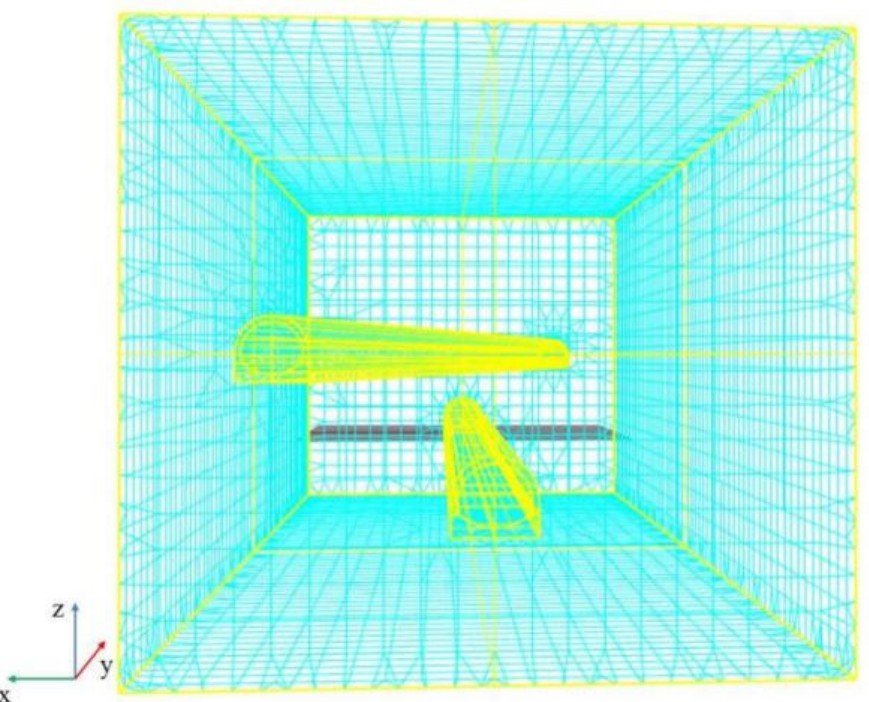

**Figure 5.** Position of crossing tunnels. (The blue grid is ground; the yellow parts are tunnels.)

The boundary conditions are depicted in Figure 6. The velocity and movement in the X and Y directions were constrained. The bottom was also restrained, while the top of the model was free. Monitoring points were laid on the surface of the model along the *y*-axis. At the position of y = 150 m (the cross-section of the two tunnels), a settlement measurement line was laid along the *x*-axis. First, we excavated Yangjiagou Tunnel in the simulation model. Since the impact of the excavation of the new tunnel on the existing one is of greater concern in this study, the Yangjiagou Tunnel was excavated using full

cross-section excavation to further simplify the calculation process. After that, monitoring points were laid on the crown, invert, and two sidewalls of the existing tunnel. Then, the tunneling process for the Donghuashan Tunnel was simulated. A three-step excavation was adopted, with the excavation increment of each step being two meters. After completing each excavation step, anchors and lining support were applied to protect the free face caused by tunneling, as shown in Table 4. Before computation, the stress condition of the model should be initialized, as shown in Figure 7.

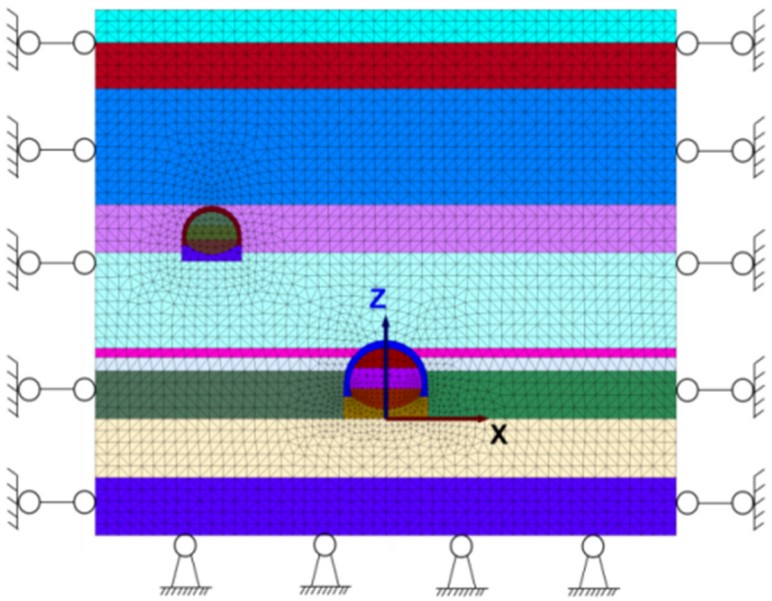

**Figure 6.** Boundary constraints.

**Table 4.** Support parameters.

| Anchor Number of Every Ring Lining | Anchor Diameter/mm | Anchor Length/m | Anchor Bond Strength/(N/m) | Anchor Space | Anchoring Method |
|---|---|---|---|---|---|
| 5 | 90 | 12 | $1.75 \times 10^5$ | 2 | Extended anchorage |

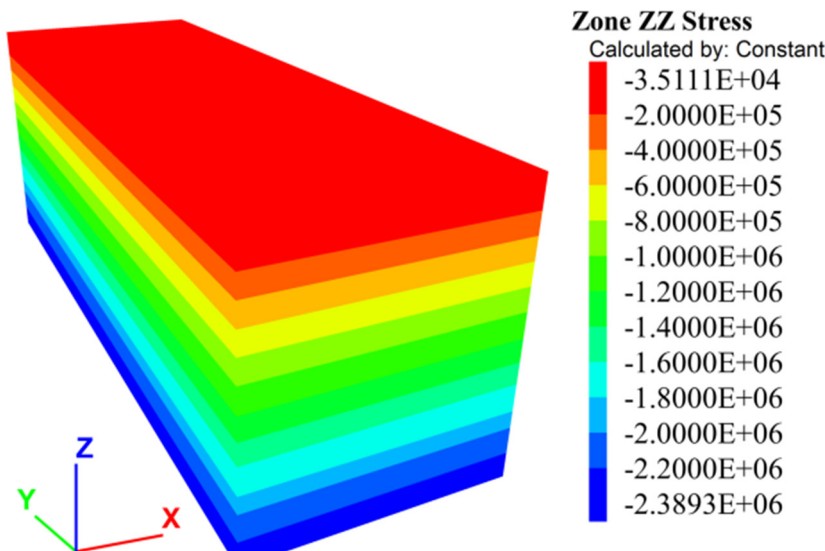

**Figure 7.** Equilibration of initial stress.

## 3. Results

### 3.1. Results of Orthogonal Tests

The numerical simulation results of the 25 groups of experiments are listed in Table 5. Here, we selected the results of the monitoring point on the ground surface above the crossing point of the two tunnels for analysis. The maximum subsidence (*W*), horizontal movement (*U*), curvature (*K*), tilt (*i*), and horizontal deformation (*ε*) are the discriminative indexes of ground movement and deformation. *W* and *U* are obtained directly from the monitoring data, while *K*, *i*, and *ε* are calculated from the surface subsidence curve, which is perpendicular to the y-axis and above the intersection zone of the two tunnels.

**Table 5.** Simulation results of orthogonal tests.

| Experiment Group | Factors | | | | | Results | | | | |
|---|---|---|---|---|---|---|---|---|---|---|
| | $\alpha/°$ | $l$/m | $h$/m | $D$/m | $\nu$ | $W$/ mm | $U$/ mm | $K$ mm/m² | $i$ mm/m | $\varepsilon$ mm/m |
| 1 | 11.3 | 13.5 | 56.58 | 14.52 | 0.7 | −13.6 | −0.432 | 0.0030 | 0.087 | 0.1002 |
| 2 | 11.3 | 10 | 50 | 12 | 0.8 | −9.88 | −0.454 | 0.0016 | 0.154 | 0.2603 |
| 3 | 11.3 | 15 | 60 | 14 | 1.0 | **−25.4** | −1.267 | 0.0053 | 0.225 | 0.1328 |
| 4 | 11.3 | 20 | 70 | 16 | 0.6 | **−27.6** | −0.634 | 0.0071 | 0.109 | 0.1327 |
| 5 | 11.3 | 25 | 80 | 18 | 0.5 | **−38.4** | −0.708 | 0.0047 | 0.110 | 0.1157 |
| 6 | 30 | 13.5 | 50 | 14 | 0.6 | −17.7 | −1.108 | 0.0018 | 0.205 | 0.1972 |
| 7 | 30 | 10 | 60 | 16 | 0.5 | **−29.5** | −0.033 | 0.0022 | 0.036 | 0.0045 |
| 8 | 30 | 15 | 70 | 18 | 0.7 | **−40.1** | **−2.698** | 0.0078 | 0.137 | 0.4346 |
| 9 | 30 | 20 | 80 | 14.52 | 0.8 | **−27.1** | **−2.281** | 0.0197 | 0.276 | 0.2679 |
| 10 | 30 | 25 | 56.58 | 12 | 1.0 | −18.9 | −1.950 | 0.0038 | 0.272 | 0.2193 |
| 11 | 50 | 13.5 | 60 | 18 | 0.8 | **−34.7** | **−3.055** | 0.0099 | 0.125 | 0.1855 |
| 12 | 50 | 10 | 70 | 14.52 | 1.0 | −14.9 | −1.792 | 0.0021 | 0.011 | 0.0341 |
| 13 | 50 | 15 | 80 | 12 | 0.6 | −13.4 | −0.874 | 0.0007 | 0.022 | 0.0296 |
| 14 | 50 | 20 | 56.58 | 14 | 0.5 | −16.7 | 0.994 | 0.0042 | 0.088 | 0.1072 |
| 15 | 50 | 25 | 20 | 16 | 0.7 | **−50.1** | **−5.611** | 0.0264 | 0.815 | 0.5950 |
| 16 | 70 | 13.5 | 70 | 12 | 0.5 | −12.5 | −1.002 | 0.0015 | 0.029 | 0.0309 |
| 17 | 70 | 10 | 80 | 14 | 0.7 | −23.4 | −1.744 | 0.0016 | 0.025 | 0.0412 |
| 18 | 70 | 15 | 56.58 | 16 | 0.8 | **−28.1** | **−2.379** | 0.0006 | 0.089 | 0.1176 |
| 19 | 70 | 20 | 50 | 18 | 1.0 | **−63.2** | **−4.914** | 0.0206 | 0.265 | 0.2988 |
| 20 | 70 | 25 | 60 | 14.52 | 0.6 | **−21.4** | −1.810 | 0.0055 | 0.042 | 0.0985 |
| 21 | 90 | 13.5 | 80 | 16 | 1.0 | **−36.1** | **−4.130** | 0.0017 | 0.041 | 0.0359 |
| 22 | 90 | 10 | 56.58 | 18 | 0.6 | **−31.2** | **−2.319** | 0.0050 | 0.098 | 0.1363 |
| 23 | 90 | 15 | 50 | 14.52 | 0.5 | −9.94 | −1.448 | 0.0011 | 0.043 | 0.0449 |
| 24 | 90 | 20 | 60 | 12 | 0.7 | −13.3 | −1.741 | 0.0001 | 0.041 | 0.0481 |
| 25 | 90 | 25 | 70 | 14 | 0.8 | −23.8 | **−3.794** | 0.0025 | 0.036 | 0.0506 |

Note: bold values could not satisfy the construction safety requirement.

According to the Technical Regulations for Monitoring and Measurement of Railway Tunnels (Q/CR9218-2015) [35], the settlement warning value is 20 mm, the allowable value is 30 mm, and the settlement rate should be no more than 5 mm/d. The critical values for horizontal deformation, tilt, and curvature are 2 mm/m, 3 mm/m, and 0.2 mm/m², respectively. The horizontal movement should be no more than 2 mm.

Figure 8 presents the ground surface settlement law during the under-crossing tunneling (the left side of the y-axis is the curve of the old tunnel excavation, and the right side is that of the new tunnel). The subsidence varies with different parameters of the newly excavated tunnel. As shown in Figure 8, the monitoring point above the intersection of the two tunnels is an inflection point for displacement during both the excavation of the old tunnel and the excavation of the new tunnel. For the excavation of the old tunnel, ground displacement experiences a rapid increase at the intersection point. After the excavation face passes that point, the increase in the settlement slows down and finally achieves stability. During the new tunnel's excavation, the original stability of the old tunnel is disturbed and the settlement of the ground surface enters a new increasing stage, which follows a similar variation law to that of the old tunnel excavation. The settlement tends to become stable after the new tunnel passes the existing one. It should be mentioned that the increase in ground subsidence at this stage is much bigger than that of the previous one. The excavation of the new tunnel induces more ground settlement, which develops on the basis of subsidence caused by previous tunneling.

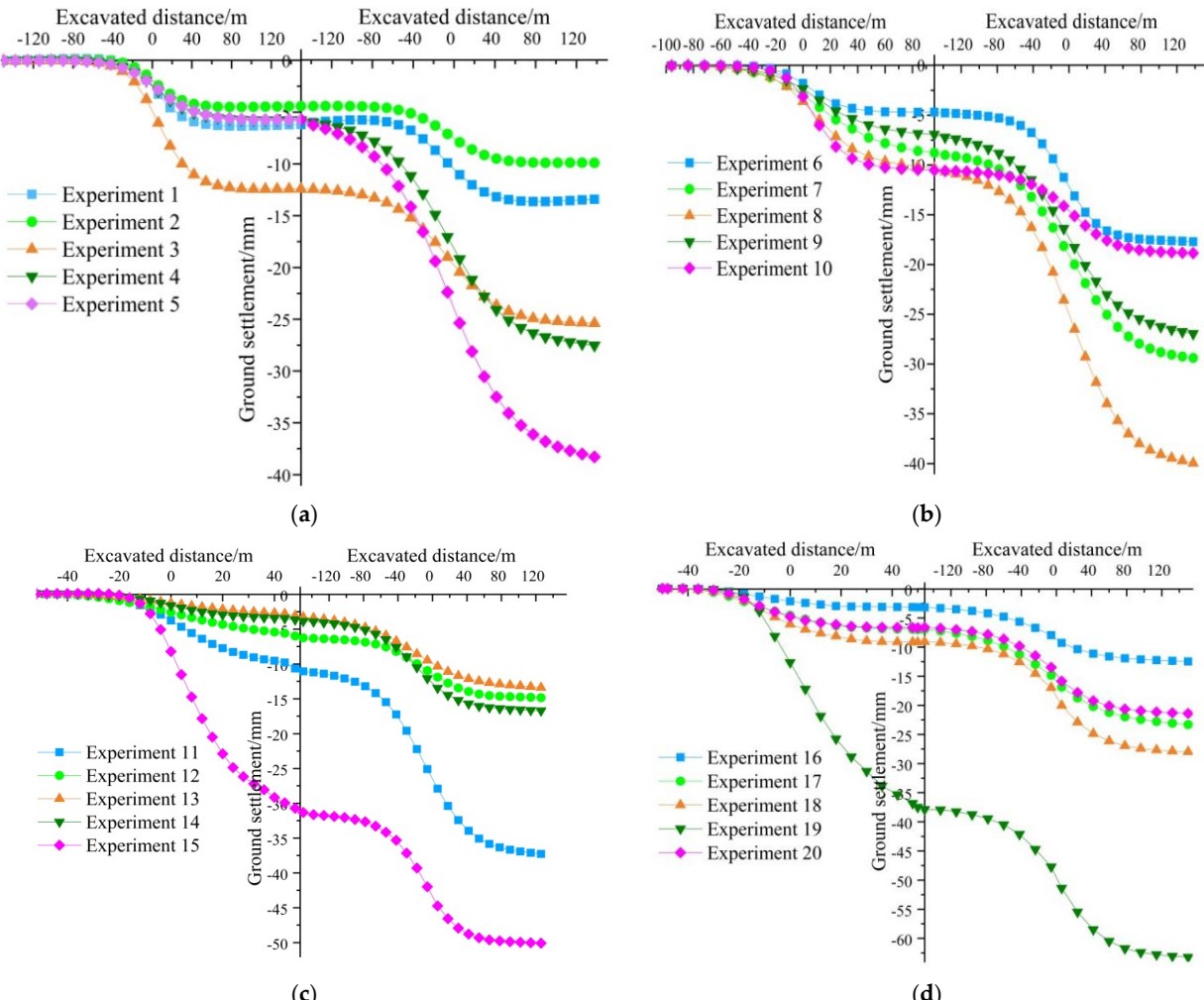

**Figure 8.** *Cont.*

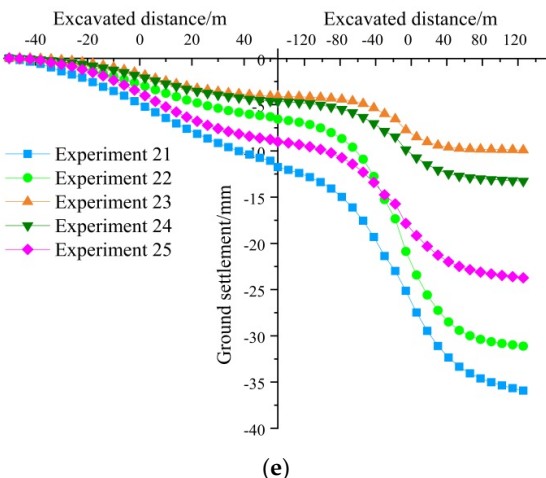

(**e**)

**Figure 8.** Ground subsidence above intersection of the two tunnels. (**a**) Experiments 1–5, (**b**) Experiments 6–10, (**c**) Experiments 11–15, (**d**) Experiments 16–20, (**e**) Experiments 21–25.

### 3.2. Results of Impact Parameter Analysis

(1)     Impact factors on ground subsidence (*W*)

According to the range analysis calculated from Table 5, tunnel clearance contributes the most to ground surface settlement, with the range *R* = 27.92. Ground subsidence varies directly with tunnel clearance, which is also reflected in Figure 9a. This means that a larger tunnel clearance will induce more ground subsidence during under-crossing tunneling. The tunnel clearance is followed by clearance ratio, proximity distance, buried depth, and skew angle. In other words, the impact of skew angle between the two tunnels on ground subsidence is the lowest. Figure 9a shows that the proximity distance and clearance ratio of the two tunnels are directly proportional, while the buried depth is inversely proportional, to surface settlement.

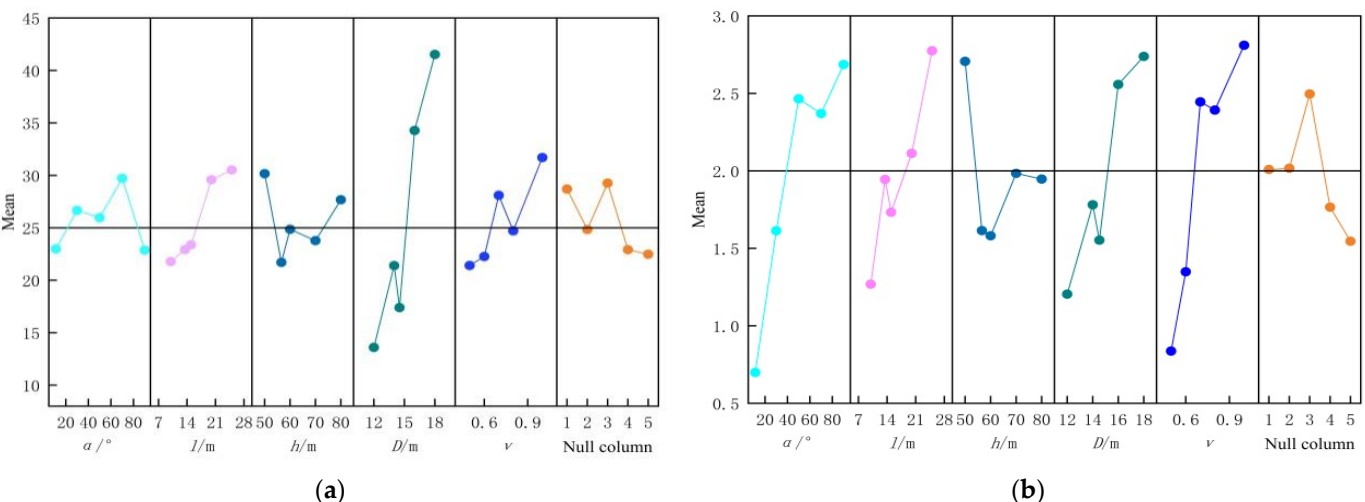

(**a**)                                                                                    (**b**)

**Figure 9.** *Cont.*

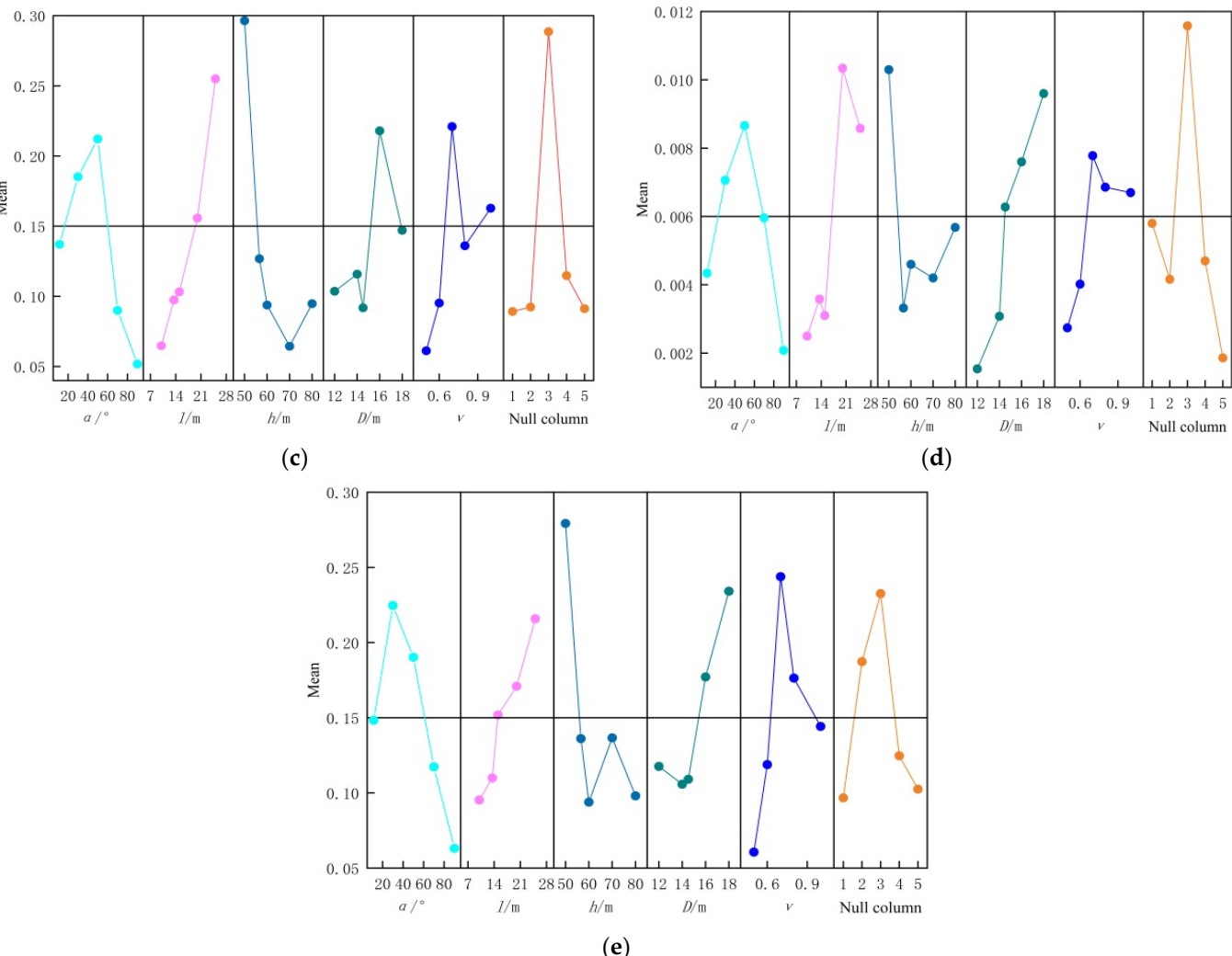

**Figure 9.** Index trend chart of impact factors on ground displacement and deformation. (**a**) Impact parameters on subsidence, (**b**) impact parameters on horizontal movement, (**c**) impact parameters on tilt, (**d**) impact parameters on curvature, (**e**) impact parameters on horizontal deformation.

(2)    Impact factors on horizontal movement (*U*)

Figure 9b presents the index trend of impact factors on horizontal movement. It is concluded that all factors except the buried depth are directly proportional to the ground horizontal movement. The ranges of the five parameters on horizontal movement are skew angle (*R* = 1.9874), clearance ratio (*R* = 1.9736), clearance (*R* = 1.5346), proximity distance (*R* = 1.5062), and buried depth (*R* = 1.1258). This means that the skew angle of the two tunnels is the master-regulator of ground horizontal movement during under-passing tunnel excavation. The larger the intersection angle between the existing tunnel and the new tunnel, the larger the horizontal movement. When the newly constructed tunnel under-passes the existing tunnel perpendicularly, the horizontal movement of the ground surface reaches the maximum. By contrast, the value of the new tunnel's buried depth makes little contribution to horizontal movement. When the depth of the new tunnel increases, the horizontal movement varies slightly.

(3)    Impact factors on tilt (*i*)

The tilt deformation of ground surface is calculated by the ratio of relative vertical movement to the horizontal distance of two adjacent monitoring points. This reflects the

grade of the ground surface subsidence basin in a certain direction. The calculation formula of $i$ is as follows:

$$i_{a-b} = \frac{W_b - W_a}{l_{a-b}} \tag{1}$$

where $i_{a-b}$ is the tilt between points a and b, mm/m; $W_a$ and $W_b$ are the vertical subsidences of points $a$ and $b$, respectively, mm; $l_{a-b}$ is the distance between $a$ and $b$, m.

From range analysis, the new tunnel's buried depth is the main impact parameter on tilt ($R = 0.2320$). Figure 9c shows that the tilt varies inversely to tunnel buried depth. Tunnel clearance is the least influential factor on ground surface tilt, whose range $R$ is 0.1262. It is worth mentioning that the tilt directly increases with the increase in proximity distance between the existing tunnel and the new tunnel. There is a turning point in the index trend chart of skew angle on tilt, where $\alpha = 50°$. When skew angle is smaller than 50°, tilt directly increases with the increase in intersection angle. After that, the $i$ varies inversely to $\alpha$.

(4)    Impact factors on curvature ($K$)

The curvature deformation of ground surface is the ratio of the difference between the tilts of two adjacent monitoring lines to horizontal distance of the middle points of these two lines, which is introduced to describe the curving of the subsidence profile. This is calculated by

$$K_{a-b-c} = \frac{i_{b-c} - i_{a-b}}{\frac{1}{2}(l_{a-b} + l_{b-c})} \tag{2}$$

where $K$ is the curvature, mm/m$^2$; $i$ is tilt, mm/m; $l$ is distance, m.

Figure 9d shows the variation law of curvature based on different influential parameters. The impact law of skew angle on curvature is similar to that on tilt. The main impact factor, tunnel clearance ($R = 0.0081$), is directly proportional to the curvature. The clearance ratio between the existing tunnel and the new tunnel is the lowest impact factor, with range $R = 0.0050$.

(5)    Impact factors on horizontal deformation ($\varepsilon$)

Horizontal deformation is calculated by

$$\varepsilon = \frac{U_b - U_a}{l_{a-b}} \tag{3}$$

where $\varepsilon$ is horizontal deformation, mm/m; $U_a$ and $U_b$ are the horizontal movements of monitoring points $a$ and $b$, respectively, mm; $l_{a-b}$ is the same.

According to range analysis, $R_h > R_v > R_\alpha > R_D > R_l$. Buried depth is the key influential factor on horizontal deformation, which is inversely proportional to $\varepsilon$ (Figure 9e). The proximity distance between the existing tunnel and the new tunnel has the least impact on horizontal deformation, and the two indexes are directly proportional. A possible cause of this phenomenon is that the impact of different parameters on the ground horizontal deformation cannot regress to a single formula. However, the general trend of impact law is not affected by this discreteness.

### 3.3. Comparison with Monitoring Data

During the construction process, settlement monitoring points were installed along the existing Yangjiagou Tunnel section from K163 + 170 to K163 + 535. Among them, settlement observation cross-sections were established every 15 m in the segments from K163 + 170 to K163 + 215 and from K163 + 475 to K163 + 535. In the segment from K163 + 215 to K163 + 475, settlement observation cross-sections were set every 10 m. Monitoring points were set at the surface level of the tunnel intersection at D1K151 + 851 to monitor ground settlement, as shown in Figure 10. On the observation cross-sections, five settlement monitoring points were installed along the tunnel's crown, left and right sidewalls, and the foot of the arch, as illustrated in Figure 11. During construction, settlement values were measured within 30 min of completing key processes such as excavation

and support, as well as lining construction, at each step of the upper, middle, and lower platforms. The monitoring frequency was once an hour.

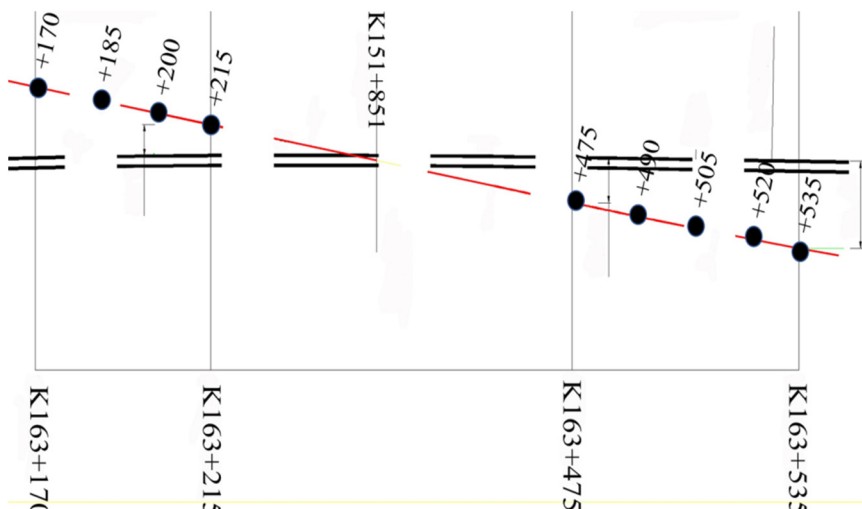

**Figure 10.** Layout of monitoring section. (The red line is the existing tunnel; the black line is the new tunnel.)

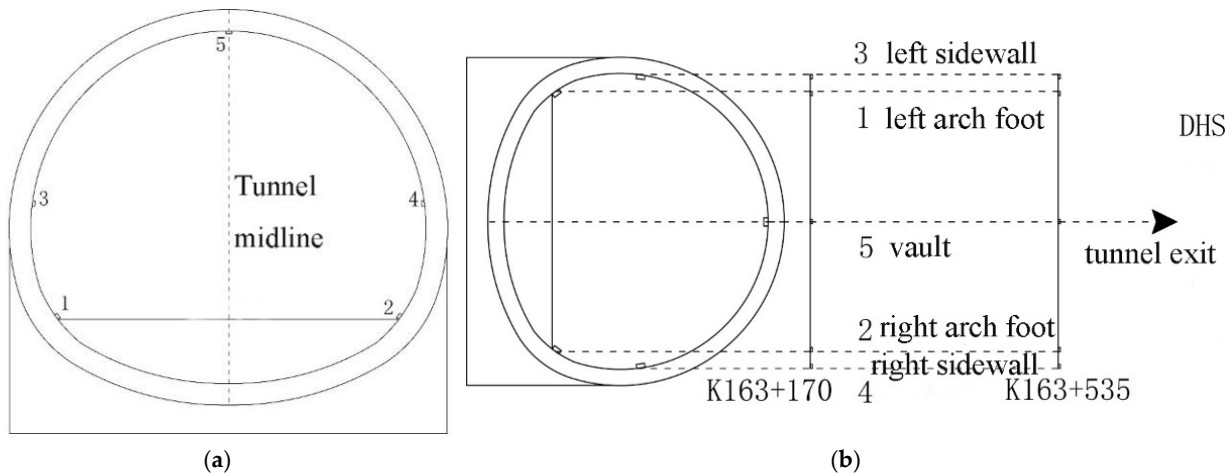

(**a**)             (**b**)

**Figure 11.** Settlement monitoring point layout. (**a**) Tunnel section, (**b**) horizontal layout.

For the convenience of research, the settlement data of five monitoring sections were taken at intervals of 15 m within the cross-influence zone from K163 + 475 to K163 + 535, and the settlement data of five monitoring sections were also taken at intervals of 26 m within the cross-core zone from K163 + 345 to K163 + 449. Along the tunnel advancement direction, starting from K163 + 535, the settlement curves of the existing tunnel during the excavation of the new tunnel and the ground settlement curves at the intersection points were plotted as shown in Figures 12 and 13, respectively. Among them, the maximum settlement value of the existing tunnel was measured at the tunnel intersection point, which is 15.96 mm, with an average settlement rate of 0.3 mm/d. The maximum settlement measured at the intersection point of the ground was 10.95 mm. The settlement was controlled by grouting with special cement, combined with control blasting+mechanical milling method in the real construction project.

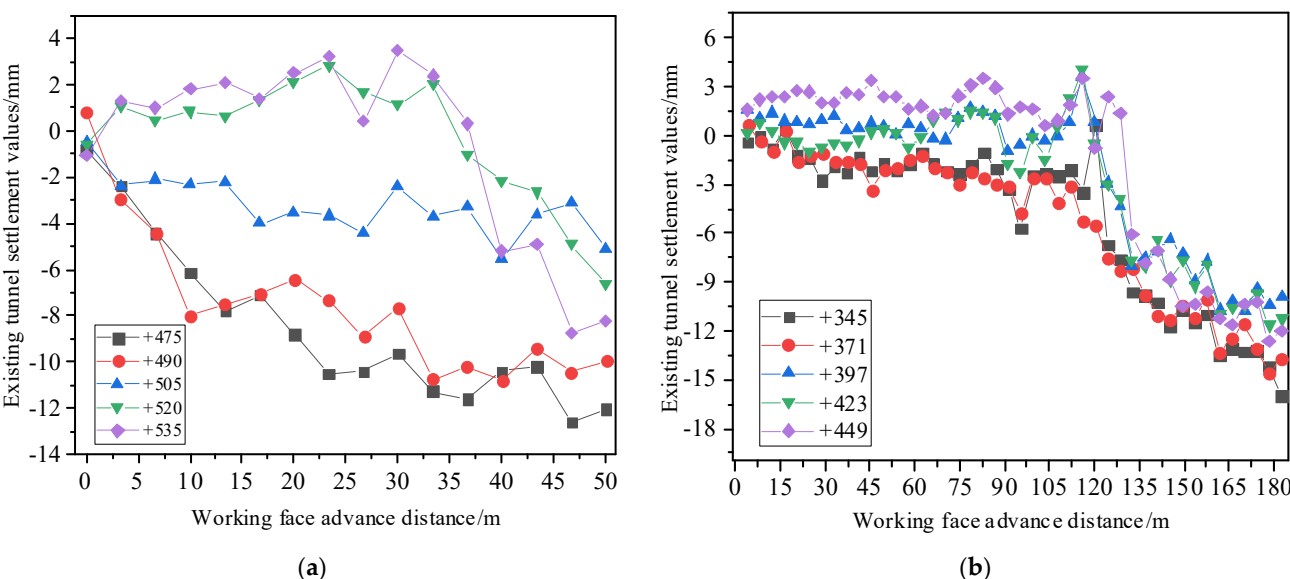

**Figure 12.** The settlement of the existing tunnel changes with the advancement of the working face. (**a**) Cross-influence zone, (**b**) intersection core area.

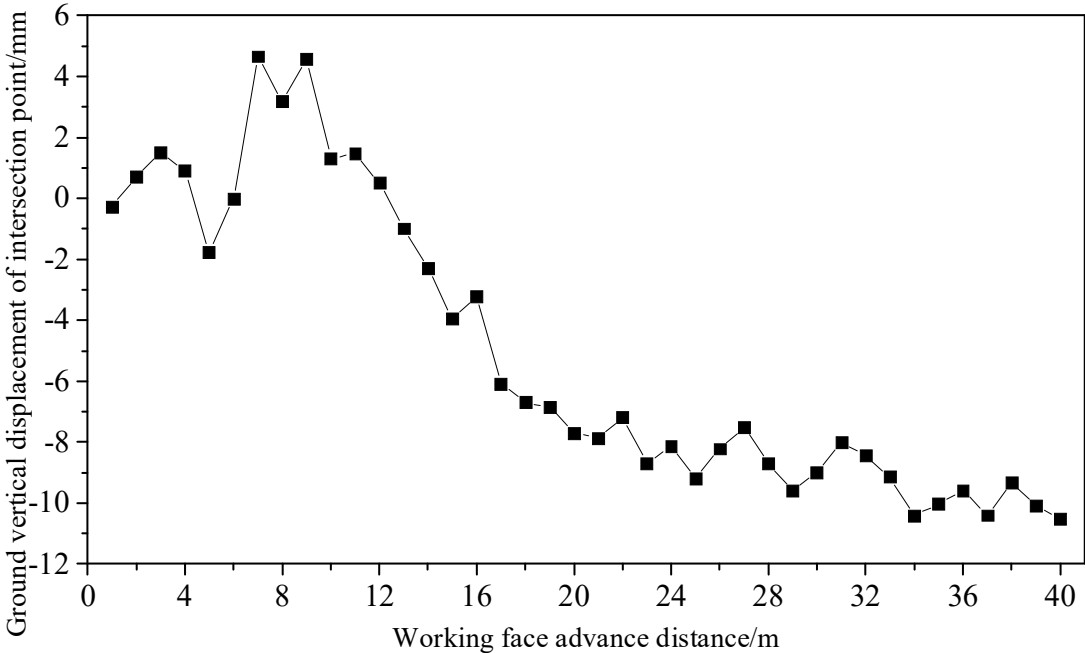

**Figure 13.** Ground settlement curve at tunnel intersection.

The first set of orthogonal tests in Table 5 is based on the prototype of Donghuashan Tunnel. The ground subsidence law of the numerical simulation is consistent with the monitoring result. Compared with the monitoring settlement of 10.95 mm, the simulation result of 13.6 mm is reliable.

## 4. Discussion

The buried depth of the new tunnel has a powerful influence on the ground movement and deformation in the construction of tunneling that under-crosses an existing tunnel. This influence affects both the key impact factor of the tilt and the horizontal deformation, indicating that there is an internal connection between these two parameters of ground movement. Most tunnel excavations in cities involve shallow tunneling, where the buried depths are generally within 10–50 m, such as shield tunnels of subways and underpass

tunnels of street roads. Tunnel excavation breaks the in situ stress field equilibrium of surrounding rocks, thus inducing stress redistribution. The Протодьяконов Arch Theory [36] indicates that when the thickness of the tunnel overlying strata is sufficient, a compressive arch will be formed to support the strata and maintain their stability. Unfortunately, it is difficult to create this kind of mechanical structure in shallow buried tunnels, where the overlying structures are more vulnerable due to the propagation of displacement and deformation induced by tunnel excavation. The aforementioned analysis reveals the role played by buried depth in ground surface settlement and deformation. A shallower excavation depth will induce more tilt and horizontal deformations in the ground surface, and vice versa.

Tunnel clearance (cross-section size of the tunnel) also plays an important role in adjacent tunnel excavation. It directly determines the ground settlement as well as the curvature. The ground subsidence is the disaster phenomenon of environmental geology with decreases in regional ground level, which is induced by the removal of underground support materials. Corresponding to this, the tunnel clearance directly contributes to the loss of supporting masses, which usually determine the extent of the ground settlement. The ground curvature deformation increases with the increase in tunnel clearance for the same reason.

By contrast, the skew angle is the primary influential parameter on ground horizontal movement. It is not difficult to understand that the ground horizontal displacement direction is perpendicular to the advancing direction of the newly excavated tunnel, which is approximately consistent with the advancing direction of the old tunnel. The horizontal movement increases with the increase in skew angle $\alpha$. When $\alpha$ reaches the maximum, that is, when the new tunnel is excavated perpendicular to the existing tunnel, the maximal horizontal movement will be induced. Figure 14 shows the impact of skew angle on horizontal movement, selected from Experiment 1 and Experiment 22 in Section 2.2. The maximum horizontal displacement increases by 17.9% when the skew angle between the existing tunnel and the new tunnel increases from 11.3° to 90°.

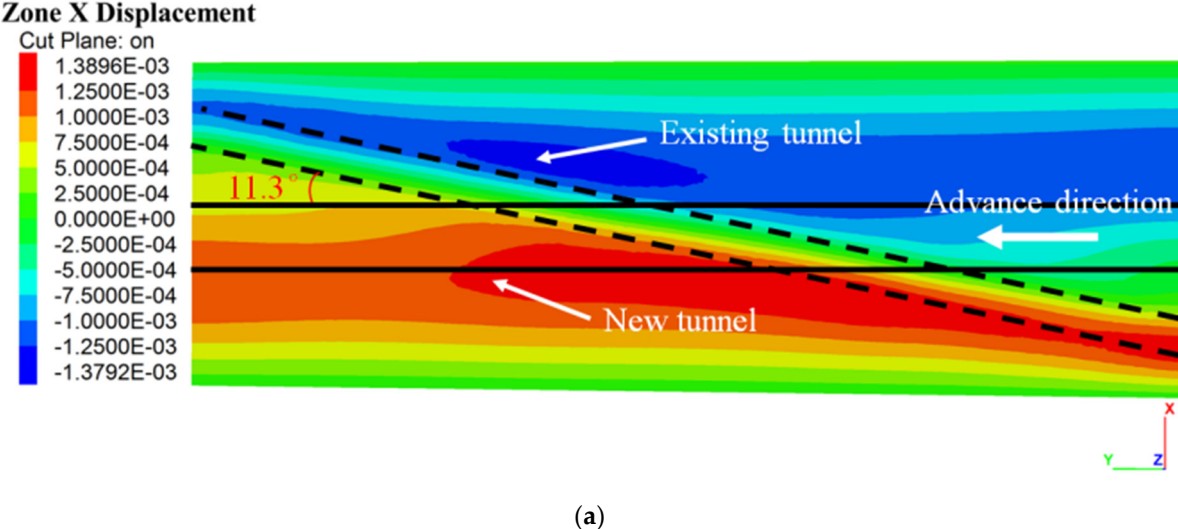

(**a**)

**Figure 14.** *Cont.*

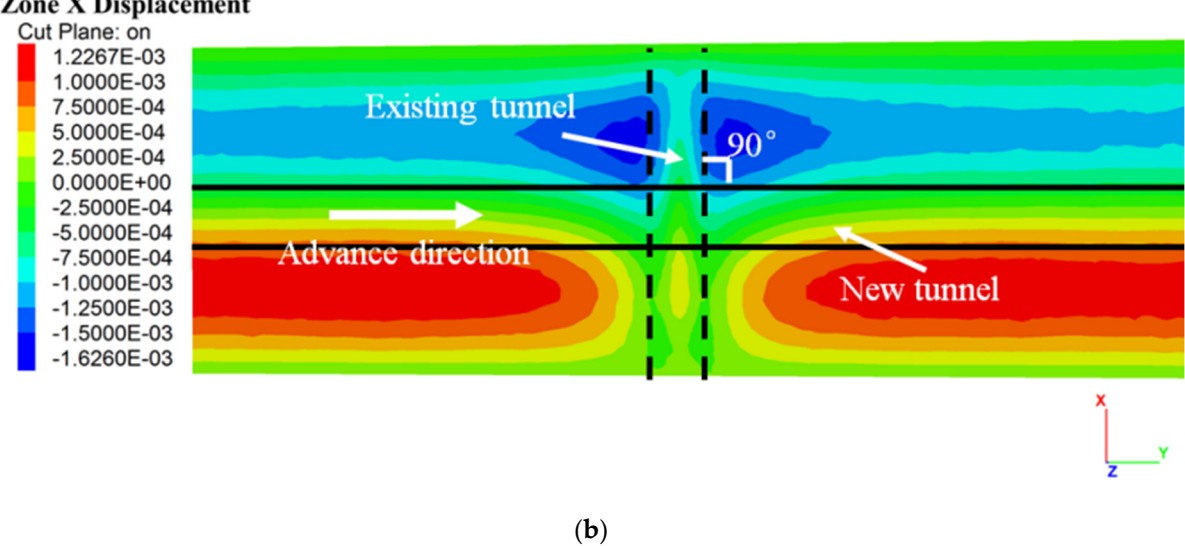

(**b**)

**Figure 14.** Horizontal displacement contour of different skew angles. (**a**) Skew angle of 11.3°, (**b**) skew angle of 90°.

## 5. Conclusions

Finite difference method (FDM) numerical analyses were carried out to evaluate the impact of different parameters on ground displacement and deformation in tunneling beneath an existing tunnel. The influential mechanism of five factors, the skew angle between the existing tunnel and the new tunnel ($\alpha$), the proximity distance of the two tunnels ($l$), the buried depth of the new tunnel ($h$), new tunnel clearance ($D$), and the clearance ratio between the two tunnels ($v$) were investigated based on 25 sets of orthogonal experimental results. Five main conclusions can be drawn from the research of this paper:

(1) Of the five influential parameters, tunnel clearance has the most significant influence on ground subsidence, while the skew angle has the smallest influence. Ground settlement directly increases with the increase in tunnel clearance.

(2) The skew angle is the main impact factor of ground horizontal movement, which should not be neglected during tunnel excavation that under-crosses existing tunnels. The horizontal movement of the ground induces shear stress along its direction, thus influencing the stability of existing tunnels. The intersection angle between the two tunnels is directly proportional to horizontal movement.

(3) The key influential parameter of ground tilt deformation and ground horizontal deformation is the new tunnel's buried depth. A shallower tunnel depth will induce larger tilt and horizontal deformation, which adversely affects the safety of structures on the ground. Proper measures should be taken to protect structures and buildings in shallow tunnel excavation projects.

(4) The ground curvature deformation is significantly influenced by new tunnel clearance. However, the clearance ratio between the two tunnels makes the smallest contribution.

**Author Contributions:** Writing—original draft, X.Y. and Y.C.; formal analysis, H.W. (Hongchao Wang); software, S.K. and C.L.; writing—review and editing, X.Z.; resources, C.W.; visualization, H.Z.; supervision, C.Y., Y.Z. and H.W. (Hua Wu). All authors have read and agreed to the published version of the manuscript.

**Funding:** This research received no external funding.

**Institutional Review Board Statement:** Not applicable.

**Informed Consent Statement:** Not applicable.

**Data Availability Statement:** No data are available.

**Conflicts of Interest:** The authors declare no conflict of interest.

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
