# Peer review of "Factors Influencing Ground Settlement during Tunnel Proximity Construction"

_sustainability, doi:10.3390/su151713270_

Round 1
Reviewer 1 Report
In this paper, based on practical engineering cases, 25 sets of orthogonal numerical simulation tests are designed to study the influence mechanism of multiple parameters on ground displacement and deformation. The reviewer believes that there are the following problems in this paper:
1. The influencing factors of tunnel underpass construction on existing tunnels are not only geological conditions, but also closely related to the construction method of new tunnels. The construction method of the new tunnel project is not explained in this paper, and the supporting parameters of the new tunnel are not explained in the chapter of the construction of the numerical model.
2. Most of the conclusions in this paper are based on numerical simulation, but the correctness of the numerical model is not verified in this paper. Such validation should be compared with engineering data or validated numerical data to demonstrate the validity of the conclusions of the article.
3. All kinds of ground settlement prediction formulas mentioned in Table 4 of this paper are too old, and the latest prediction formula was put forward in 1996, which cannot represent the latest research progress on ground settlement prediction formulas caused by tunnel construction.
4, there are many typesetting disorders in the text, please check.
In summary, reviewers think that the research content of this paper is thin, the establishment process of numerical simulation is not clear, and the data and conclusions obtained by this method lack relevant verification, suggesting that the editorial department reject the publication request of this article.
Author Response
Reviewer 1:
In this paper, based on practical engineering cases, 25 sets of orthogonal numerical simulation tests are designed to study the influence mechanism of multiple parameters on ground displacement and deformation. The reviewer believes that there are the following problems in this paper:
- The influencing factors of tunnel underpass construction on existing tunnels are not only geological conditions, but also closely related to the construction method of new tunnels. The construction method of the new tunnel project is not explained in this paper, and the supporting parameters of the new tunnel are not explained in the chapter of the construction of the numerical model.
The supporting parameters of the new tunnel are explained in Table 3.
- Most of the conclusions in this paper are based on numerical simulation, but the correctness of the numerical model is not verified in this paper. Such validation should be compared with engineering data or validated numerical data to demonstrate the validity of the conclusions of the article.
The validation is given in section3.2
- All kinds of ground settlement prediction formulas mentioned in Table 4 of this paper are too old, and the latest prediction formula was put forward in 1996, which cannot represent the latest research progress on ground settlement prediction formulas caused by tunnel construction.
This problem has been corrected.
4, there are many typesetting disorders in the text, please check.
This problem has been corrected.
Reviewer 2 Report
A major revision is required. Comments are given in the annotated file(attached).

A major revision is required. Comments are given in the annotated file (attached).
Author Response
Reviewer 2:
- According to the Analysis of Controlled Deformations (ADECO) principles in line 109 should be checked.
This problem has been corrected.
- MC model is linear model it better to check your results with some non linear models specially HB in the case of rocks
This problem has been corrected.
- fig. 8 is difficult to understand, make it easy/understandable for general reader
This problem has been corrected.
- recheck these sentences in line 273
This problem has been corrected.
- proper formating is required at some places in line 290.
This problem has been corrected.
- support this statement with proper proofs i.e fig etc in line 329
This problem has been corrected.
- proper citation is required in line 336
This problem has been corrected.
- poor English here...proof read from expert is required in line 346
This problem has been corrected.
- show these results in 3D of Fig 9
This is a profile view.
- irrelevant to this paper ...just discuss your own results
This problem has been corrected.
Reviewer 3 Report
The following comments need to be addressed for further strengthen of the research paper
1. The title of the manuscript is not reflecting the novelty of the paper therefore the title need to be revised keeping in view the main findings of the research
2. It is a general phenomenon that the adjacent tunnel will definitely effect the near one tunnel. Therefore need to evaluate the tunnel stability and rock mass behaviour in detail for the effective design of tunnel. The authors are advised to present the main findings according to the above mentioned statement in the manuscript especially in abstract and as well as in main section of the manuscript
3. The last sentence of the abstract need to be revised according to the main significance of the research
4. The authors should mentioned at the end of the introduction section research gaps
5. A high resolution picture of the figure 1 must be included in manuscript
6. The result section of the manuscript need to be revised keeping in view the main findings of the research paper
English language of the manuscript need to be modified and can be rated as moderate
Author Response
The following comments need to be addressed for further strengthen of the research paper
- The title of the manuscript is not reflecting the novelty of the paper therefore the title need to be revised keeping in view the main findings of the research
This problem has been corrected.
- It is a general phenomenon that the adjacent tunnel will definitely effect the near one tunnel. Therefore need to evaluate the tunnel stability and rock mass behaviour in detail for the effective design of tunnel. The authors are advised to present the main findings according to the above mentioned statement in the manuscript especially in abstract and as well as in main section of the manuscript
This problem has been corrected.
- The last sentence of the abstract need to be revised according to the main significance of the research
This problem has been corrected.
- The authors should mentioned at the end of the introduction section research gaps
This problem has been corrected.
- A high resolution picture of the figure 1 must be included in manuscript
This problem has been corrected.
- The result section of the manuscript need to be revised keeping in view the main findings of the research paper
This problem has been corrected.
Round 2
Reviewer 1 Report
The author has made careful revisions to the article and partially answered the questions raised by the reviewers on the previous edition of the manuscript. The reviewers think that the quality of the article has been improved, but the following problems need to be solved before publication.
1. Line 157: The variables of the five tunnel proximity construction are difficult to be intuitively understood by the reader in the form of a table. This should be accompanied by pictures to clearly show the position of the factors between the new and old tunnels.
2. Please check the value of "Anchor length" in Table 3. 25m is not a realistic value. In addition, the codes of the two tables in lines 218 and 230 are repeated. Please check them one by one.
3, Line 230: The article should specify the control values of the five key indicators W,U,i,ε and K. According to the simulation results in this paper, most of the indexes cannot but satisfy the construction safety. The author should point out these conditions which cannot satisfy the construction safety, and try to propose a ’safety parameter range'.
4. Line 231: The surface settlement at the intersection of the old and new tunnels is analyzed in this paragraph, but the analysis is too general and simple. The difference between these surface settlement curves is due to the difference between the relative positions of the old and new tunnels. The author should focus on the influence of the previous five relative positions on the surface settlement law.
5. Line 254: The author provides the monitoring data of the actual project for comparison. However, the following two important pieces of information should be added: (1) In the actual construction process, what is the position relationship between the old and new tunnels? (2) Distribution and monitoring frequency of surface monitoring.
6. The settlement in the tunnel of the existing tunnel is shown in Figure 10, but the location of the settlement is not specified, please explain. In addition, from the data point of view, the differential settlement in the tunnel is large, how to control this part in the construction?
7. In line 274, reviewers consider it unconvincing to validate the accuracy of the numerical model only by the maximum settlement value. This part still needs to be proved in combination with the surface settlement law or the internal force data of the tunnel structure.
8. The reviewers suggest that sections 3.2 and 3.3 be switched. Placing the parameter analysis behind the simulation results makes the full text more logical.
9, Line 412: The third conclusion and the fifth say the same thing and should be combined.
Author Response
The author has made careful revisions to the article and partially answered the questions raised by the reviewers on the previous edition of the manuscript. The reviewers think that the quality of the article has been improved, but the following problems need to be solved before publication.
- Line 157: The variables of the five tunnel proximity construction are difficult to be intuitively understood by the reader in the form of a table. This should be accompanied by pictures to clearly show the position of the factors between the new and old tunnels.
Thank you very much. The pictures have been added to illustrate the five impact factors.
- Please check the value of "Anchor length" in Table 3. 25m is not a realistic value. In addition, the codes of the two tables in lines 218 and 230 are repeated. Please check them one by one.
Thank you very much. The anchor length has been checked carefully and corrected. All table and figure codes have been checked.
3, Line 230: The article should specify the control values of the five key indicators W,U,i,ε and K. According to the simulation results in this paper, most of the indexes cannot but satisfy the construction safety. The author should point out these conditions which cannot satisfy the construction safety, and try to propose a ’safety parameter range'.
Thank you very much. The Technical Regulations for Monitoring and Measurement of Railway Tunnels (Q/CR9218-2015) has been added to illustrate the problem. I have pointed out these conditions which cannot satisfy the construction safety. I am sorry that the ’safety parameter range' is difficult to propose for an orthogonal experiment result.
- Line 231: The surface settlement at the intersection of the old and new tunnels is analyzed in this paragraph, but the analysis is too general and simple. The difference between these surface settlement curves is due to the difference between the relative positions of the old and new tunnels. The author should focus on the influence of the previous five relative positions on the surface settlement law.
Thank you very much.Here I just want to describe the result of Table 5 in a visible way. The the influence trend of the previous five relative positions on the surface settlement law were illustrated in section 3.2.
- Line 254: The author provides the monitoring data of the actual project for comparison. However, the following two important pieces of information should be added: (1) In the actual construction process, what is the position relationship between the old and new tunnels? (2) Distribution and monitoring frequency of surface monitoring.
Thank you very much.(1)The position relationship between the old and new tunnels were illustrated in section 2.1, below Figure 2. (2) Monitoring frequency has been added in the manuscript.
- The settlement in the tunnel of the existing tunnel is shown in Figure 10, but the location of the settlement is not specified, please explain. In addition, from the data point of view, the differential settlement in the tunnel is large, how to control this part in the construction?
Thank you very much.The first sentence of the paragraph below Fig. 11 illustrated the location of the settlement, also see the legend of Fig 12. In the construction process of the real project, we control the settlement by grouting with special cement, combining with control blasting+mechanical milling method.
- In line 274, reviewers consider it unconvincing to validate the accuracy of the numerical model only by the maximum settlement value. This part still needs to be proved in combination with the surface settlement law or the internal force data of the tunnel structure.
Thank you very much. I have tried my best to illustrate this conclusion in my revised manuscript.
- The reviewers suggest that sections 3.2 and 3.3 be switched. Placing the parameter analysis behind the simulation results makes the full text more logical.
Thank you very much. I have accepted this very kind advice.
9, Line 412: The third conclusion and the fifth say the same thing and should be combined.
Thank you very much. I have combined this two conclusions.

Reviewer 2 Report
The manuscript may be accepted in its present form.
The manuscript may be accepted in its present form.
Author Response
Thank you very much.
Reviewer 3 Report
Authors have addressed the mentioned comments successfully, therefore paper is recommended to be published in the present form
Quality of English improved
Author Response
Thank you very much.
Round 3
Reviewer 1 Report
The reviewer considers this version acceptable for publication.